# Water versus Oil Lubrication of Laser-Textured Ti6Al4V Alloy upon Addition of MoS_2_ Nanotubes for Green Tribology

**DOI:** 10.3390/ma15092974

**Published:** 2022-04-19

**Authors:** Marjetka Conradi, Bojan Podgornik, Aleksandra Kocijan, Maja Remškar, Damjan Klobčar

**Affiliations:** 1Institute of Metals and Technology, Lepi Pot 11, 1000 Ljubljana, Slovenia; bojan.podgornik@imt.si (B.P.); aleksandra.kocijan@imt.si (A.K.); 2Jozef Stefan Institute, Jamova 39, 1000 Ljubljana, Slovenia; maja.remskar@ijs.si; 3Faculty of Mechanical Engineering, University of Ljubljana, Aškerčeva 6, 1000 Ljubljana, Slovenia; damjan.klobcar@fs.uni-lj.si

**Keywords:** green tribology, surface texturing, MoS_2_ nanotubes, lubrication, Ti6Al4V alloy

## Abstract

A Nd-YAG laser was used for texturing the Ti6Al4V surface with dimples of diameter 50 and 100 µm and centre-to-centre distance 100, 200 and 400 µm, defining the surface texture density. The tribological evaluation was conducted to analyse and compare the behaviour of un-textured and laser-textured samples under water in comparison to oil (PAO6) lubrication without and with the addition of MoS_2_ nanotubes into the lubricant. MoS_2_ nanotubes had a positive effect on friction in both media for laser-textured Ti6Al4V. Evaluation of friction and wear in water and PAO6 showed a comparable tribological response in water to oil for specific laser-textured configurations, proving the novel concept of green tribology for laser texturing in combination with MoS_2_ nanotubes/water lubrication.

## 1. Introduction

The alpha-beta titanium alloy Ti6Al4V is known as a lightweight, high-strength and high-performance material used in various industrial applications, such as automotive, aerospace as well as medical applications [1]. In addition to high strength and low density, its main advantages are also good corrosion resistance, biocompatibility, ductility and performance at elevated temperatures [1,2]. In spite of mentioned superior properties, the main challenge of Ti6Al4V alloy is its limited use in tribological applications due to a high coefficient of friction (COF) and poor wear resistance [3].

These drawbacks can be significantly minimised by using appropriate lubricants that improve the sliding conditions at the contact area by reducing the contact pressure [4,5,6,7]. This leads to a reduction in wear and the volume of wear debris. The most common is liquid lubrication, which has been successfully used in different applications already for several years [8,9,10]. The leading role in this field belongs to oil-based lubrication with an emphasis on low viscosity oils [11]. Another promising and rapidly developing area presents water-based lubrication, which is within the concept of green tribology focusing on the reduction in waste and elimination of hazardous substances [12]. In order to additionally improve the tribological behaviour in demanding industrial environments, the researchers also focused on the implementation of specific lubricant additives [1]. Several organic and/or inorganic nanomaterials (i.e., graphene, boron nitride, MoS_2_, ionic liquids, hydrogels, etc.) were proposed to serve as lubricant additives resulting in enhanced sliding efficiency and durability [13,14].

Another important aspect of the tribological response is the ability of lubricant retention [15]. Surface modification by laser texturing was introduced to adjust the surface characteristics of the base material to successfully overcome its limitations of use via different surface topographies by manipulating processing parameters [6,16,17,18,19,20,21]. It was already shown that the texturing geometry and direction of texturing with regard to sliding during application significantly influence the lubrication efficiency and wear behaviour [22,23]. Various texturing geometries, such as micro-channels and dimples of different sizes and depths, have already been established as traps for wear debris and lubricant reservoirs, leading to reduced abrasion [24,25]. 

In the current research, we have focused on the improvement of the tribological response of Ti6Al4V alloy by a combination of surface modification via laser texturing and lubrication. We studied the effect of the dimple-textured surface with specific dimple diameter and centre-to-centre distance on friction and wear in two configurations, along texturing direction, 0° and at the angle 45° between the direction of dimples and direction of sliding, and under four lubrication regimes: water-based without/with MoS_2_ nanotubes and oil-based without/with MoS_2_ nanotubes. Our challenge was to find the balance between the appropriate surface texture and lubrication in order to improve the tribological characteristics of Ti6Al4V alloy. Furthermore, we examined the possibility of adequately replacing oil-based lubrication with water upon the addition of MoS_2_ nanotubes with respect to frictional and wear performance. The application of water-based lubrication with MoS_2_ nanotubes is within the concept of green tribology, which focuses on balancing the tribological and ecological aspects with minimal environmental and biological impact. This offers the possibility to implement Ti6Al4V alloy in more tribologically challenging working environments.

## 2. Materials and Methods

Materials―Ti6Al4V alloy sheet (1.5 mm thickness), grade 5, 6% aluminium, 4% vanadium, in the solution-and-aged condition (37 HRC) was used as the substrate. In order to achieve the appropriate mechanical properties, the titanium alloy sheet was heat-treated for 1 h at 830 °C (±10 °C), the hold time was 1.5 h, then it was allowed to cool in the furnace. The resulting hardness was 340 HV0.5. Prior to the laser surface texturing, the alloy sheet was cut into discs of 25 mm diameter. The surfaces of these test coupons were hand-ground, using grinding paper of 600 grit to remove the surface oxide layer before processing. They were then cleaned using cotton wool and isopropanol to remove surface impurities. The average surface roughness of the prepared test coupons was S_a_ = (0.189 ± 0.01) µm. 

The NEXBASE 2006 polyalphaolefin 6 (PAO6) oil, supplier (Neste Oil, Espoo, Finland), was used for lubricated tribological testing. The kinematic viscosity of the PAO6 oil is 29.0–30.5 mm^2^/s at 40 °C and 5.7–6.0 mm^2^/s at 100 °C.

The MoS_2_ nanotubes were prepared by sulfurisation of Mo_6_S_2_I_8_ nanowires (Nanotul Ltd., Ljubljana, Slovenia) in H_2_S (1%)/H_2_ (1%)/argon (98%) flowing gas at 1073 K. The reaction run for 2 h. The product is a pure MoS_2_ composed of a mixture of nanotubes, which kept the outer shape of the starting material, and onion-like particles, which were created during the sulfurization by self-closure of thin MoS_2_ flakes. The length of the nanotubes was up to 5 μm, while the diameter was smaller than 200 nm, which is also the maximum diameter of the onions. The MoS_2_ nanotubes and nano-onions crystallise mostly in the 2H polytype (JCPDS No. 77-1716), while the presence of the 3R polytype (JCPDS No. 74-0932) cannot be completely excluded. The (103) peak of the 2H-phase is widened by the presence of the (104) and (015) peaks of the 3R polytope, while the (105) peak is widened by the (107) and (018) peaks of the 3R phase. The X-Ray diffraction (XRD) spectrum and transmission electron microscopy (TEM) image of MoS_2_ nanotubes are presented in Figure 1. The XRD spectrum was recorded with an AXS D4 Endeavor diffractometer (Bruker Corporation, Karlsruhe, Germany), with Cu Kα1 radiation and an SOL-X energy-dispersive detector with the angular range of 2θ from 5° to 75° with a step size of 0.04° and a collection time of 3 to 4 s. High-resolution transmission electron microscopy (TEM) image was acquired with Cs probe-corrected TEM/STEM Jeol ARM 200CF microscope, operated at 200 kV.

Prior to tribological testing, 2 wt.% of MoS_2_ nanotubes were dispersed in both media, water and PAO6. The wt.% of added MoS_2_ nanotubes was chosen according to the previous work [26]. The mixture was alternately put into ultrasound for 20 min and then manually shaken three times.

Surface laser texturing―A Starmark laser texturing machine was used for the production of samples. A Rofin SMD 50 W II Nd-YAG laser power source and F-Theta-Ronar lens with a focusing length of 160 mm were used for processing. The laser’s focal point was on the surface of the test samples. Programming of processing texture was performed using Rofin LaserCAD software, where a set of 50 µm and 100 µm dimples were arranged in a square with a centre-to-centre distance of 100 µm, 200 µm and 400 µm. The target depth of dimples was set to 20 µm. Laser processing was used in CW mode with a pulse length of 0.2 ms, frequency of 500 Hz, pulse spacing of 0.1 mm, and minimal pulse spacing of 0.05 mm. The 50 µm dimples were processed with 17 pulses and an electrical current of 37.0 A, and the 100 µm dimples with 11 pulses and an electrical current of 41.0 A. The laser processing was performed in air at room temperature. The times needed for processing the test coupons surfaces were: 220 s for 100 µm dimples with 400 µm spacing, 467 s for 100 µm dimples with 200 µm spacing, 736 s for 50 µm dimples with 200 µm spacing, and 2508 s for 50 µm dimples with 100 µm spacing. 

Prior to tribological testing, all the laser-textured surfaces were hand-ground with grinding paper of 1200 grit followed by grinding paper of 2400 grit to remove the bulges of the ejected material caused by laser texturing. Figure 2 shows an example of the morphology of the dimples’ textured surface immediately after laser texturing and after grinding.

Surface characterisation―A field-emission scanning electron microscope SEM JEOL JSM-6500F (SEM) was employed to analyse the morphological characteristics of the laser-textured Ti6Al4Vsurfaces. An Optical 3D metrology system, model Alicona Infinite Focus (Alicona Imaging GmbH), and IF-MeasureSuite (Version 5.1) software were used for profile measurements, wear track width measurements and wear volume calculation.

Tribological testing―Tribological testing using a ball-on-flat contact configuration was performed under a reciprocating sliding motion on a TRIBOtechnic friction testing tribometer. Friction and wear tests were performed at a normal load of 5 N, corresponding to the nominal contact pressure of 600 MPa, average sliding speed of 5 mm/s (frequency of 0.25 Hz and stroke of 10 mm), and total sliding time of 200 sec (sliding distance of 1 m). These specific contact conditions were selected in order to simulate medium-load, low sliding speed conditions, which is critical in terms of boundary lubrication. Furthermore, the main focus was on initial running-in performance. A 100Cr6 ball (750 HV) with a diameter of 10 mm was used as a counter-body, loaded by a death-weight against a moving Ti6Al4V disc. Sliding orientation was parallel with the texturing pattern (0°) as well as at a 45° angle (Figure 3). Before each test, both contact bodies were ultrasonically cleaned in ethanol, dried in the air and mounted, and a drop of lubricant was applied on the disc surface, thus resulting in boundary/mixed lubrication. Each test was repeated at least three times, and the average steady-state coefficient of friction (last 50 s) was calculated and wear volume measured with the optical 3D measurement system Alicona InfiniteFocus G4, subtracting the original textured surface from the wear track.

## 3. Results and Discussion

### 3.1. Surface Morphology

The T6Al4V discs were textured with Nd-YAG laser and then hand ground, first with 1200- and finally with 2400-grit grinding paper to remove the hardened bulges of the ejected material caused by laser texturing. As shown in our previous work [27], bulges can have a negative effect on friction and wear of the textured Ti6Al4V alloy.

The surface morphology was defined with laser parameters (power, frequency, speed, repetitions) which were chosen to produce dimples with the target depth of 20 µm and with two different diameters, 50 µm and 100 µm. As shown in Figure 4, the final texture was characterised by a square-like configuration of dimples with centre-to-centre distances of 100 µm and 200 µm for 50 µm-size dimples (50–100 and 50–200) and centre-to-centre distances of 200 µm and 400 µm for 100 µm-size dimples (100–200 and 100–400). The dimple profile analysis in Figure 4 shows that the final depth of the dimples ranged between 20 and 27 µm and that the hand grinding removed the bulges successfully.

The morphological differences between the four different surface configurations were evaluated according to their surface texture density. The texture density was calculated as the ratio between the laser-textured area (dimples) and the total surface area as observed by SEM. The 50–100 surface had the highest texture density of around 40%, the 100–200 surface of 30%, while 50–200 and 100–400 surfaces had significantly lower texture densities, around 15% and 10%, respectively. 

### 3.2. Tribological Testing

#### 3.2.1. Coefficient of Friction

Under water lubrication, the non-textured Ti6Al4V alloy showed relatively stable friction during the 200 s sliding test, with the initial and steady-state coefficient of friction of about 0.40 (Figure 5a and Figure 6a). On the textured surfaces lubricated with water, friction only slightly decreased and varied with the texture density. The lowest and the most stable coefficient of friction of ~0.35 was obtained when using 100 µm diameter dimples and 200 µm dimple-to-dimple distance (100–200; texturing density of 30%). 

Replacing water with PAO6 base oil caused, on the non-textured Ti6Al4V alloy, an increase in the coefficient of friction to about 0.45, as shown in Figure 5a and Figure 7a. Although PAO6 oil has 30 times higher viscosity than water, it wets the Ti6Al4V surface better (θ_PAO6_ = 5° vs. θ_water_ = 50°). Thus, the negative effect of higher viscosity on friction is strongly diminished. Surface texturing also reduced friction for PAO6, with the lowest value of 0.36 obtained on the 50–200 surface. 

In water, a variation in friction with the size of the dimples and the distance between them can be explained by the change in micro-hydrodynamic effects of the dimples, thus resulting in up to a 10% increase in friction when the surface is processed with small dimples with large dimple-to-dimple distance [28,29]. This is visible in an increase in CoF on the 100–400 surface with regard to the 100–200 surface. The sliding direction had no influence on the stability or the value of the coefficient of friction due to the very low viscosity of the water, as shown in Figure 5a and Figure 6.

In the case of PAO6 oil lubrication, surface texturing reduced CoF by up to 20% (Figure 5a). In this case, besides texturing density, the sliding direction also matters (Figure 7). For the sliding along the texturing pattern, the lowest and the most stable friction (~0.35) was obtained on the 50–200 surface with a texturing density of 15% (Figure 7a), while for sliding at a 45° angle, the lowest and the most stable friction was found for the 100–200 case with the 30% texturing density (Figure 7b). In this case, the wear track texturing density, defined as the number of dimples acting within the wear track, was around 27%, similar to the 50–200 at 0°.

An addition of 2 wt.% of MoS_2_ nanotubes into the lubricant had a positive effect on friction reduction, as shown in Figure 5b, indicating their action as a solid lubricant additive. The addition of MoS_2_ nanotubes did not provide a stable friction-reducing effect only in the case of non-textured surface and water lubrication, with a coefficient of friction fluctuating between 0.35 and 0.4 during the course of sliding (Figure 8a). This can be explained by agglomeration of the nanotubes, which causes their accumulation and pushing out of the contact [30,31], which results in a loss of their lubricating effect. However, for textured surfaces, dimples may act as MoS_2_ nanotubes reservoirs, providing a permanent friction-reducing effect without agglomeration problems. In the event of nanotubes agglomeration, the agglomerates are not removed from the contact but become stuck in the dimples, thus retaining their lubricating and friction reduction effects. However, the dimples’ effectiveness depends on the texturing pattern and density. The largest texturing density of 40% provided the lowest and the most stable CoF, which is reduced down to 0.15 (50–100 at 45°), as shown in Figure 5b and Figure 8. For other texturing densities, the CoF was in the range of 0.30–0.32, which is around 10–15% lower as compared to that in pure water.

An even bigger friction-reducing effect provided by MoS_2_ nanotubes was obtained when used in combination with PAO6 oil (Figure 5b and Figure 9). In this case, friction reduction of more than 30% was observed already for untextured surfaces, with the coefficient of friction being in the range of 0.25–0.30. However, surface texturing of the Ti6Al4V disc had only limited added effect in this lubricant. It provides a substantial reduction in coefficient of friction, down to 0.15, but only when using high texturing densities of over 40% (50–100 at 45° and 0° and 100–200 at 45°), as shown in Figure 5b and Figure 9. The reason is that MoS_2_ nanotubes agglomeration is largely omitted in PAO6 oil. Therefore, the main function of the dimples is not to remove agglomerates from the contact but to act as oil and nanotubes supply reservoirs and to provide micro-hydrodynamic effects. As shown by the results, the beneficial action of texturing and constant supply of nanotubes into the contact is limited to large texturing densities, mainly due to increased viscosity caused by the addition and homogeneous dispersion of MoS_2_ nanotubes [32].

#### 3.2.2. Wear

In terms of wear, no measurable wear could be detected on the 100Cr6 ball, which is more than two times harder than the Ti6Al4V disc, regardless of the texturing pattern and lubricant used. However, for the Ti6Al4V disc, water lubrication resulted in more than two times higher wear as compared to PAO6 oil, as shown in Figure 10a. This is mainly related to water’s lower viscosity and very thin lubrication film. Although the surface texturing provides a small friction-reducing effect under water lubrication, this does not reflect the reduction in wear. Contrary, the highest texturing density (50–100) even intensifies the abrasive wear (Figure 10a and Figure 11), increasing it by up to five times due to increased contact stress within the contact [33]. The addition of MoS_2_ nanotubes in water, in general, has only a minor effect on wear (Figure 10b), mainly due to their agglomeration tendency and filling up the dimples (Figure 12) when sliding is performed along the texturing direction. However, under sliding at a 45° angle and altering textured/non-textured area within the contact (Figure 13b), MoS_2_ nanotubes captured within the dimples provide a wear-reducing effect under water lubrication. Wear volume is reduced by 40–50% due to more even distribution of bearing and lubricating areas within the sliding contact (Figure 13).

Similar behaviour and abrasive wear being the dominant wear mechanism (Figure 11) was also observed for PAO 6 oil lubrication conditions, albeit at a lower degree of wear (Figure 10). This is related to higher oil viscosity and thicker lubricating film, separating contact surfaces more efficiently. Surface texturing intensified wear, especially for 50–100 cases, while the addition of MoS_2_ nanotubes in general reduced wear due to their efficient spread of the nanotubes in oil (Figure 14) and uniform flow of nanotubes into the contact. Again, the positive effect of nanotubes is stronger for 45° sliding when dimples were alternately passing through the contact (Figure 13), as shown in Figure 10. However, for 45° sliding, texturing density within the wear track is increased, resulting in higher contact stress and consequently in about 25% higher wear. 

## 4. Conclusions

This study shows that laser texturing reduces the CoF of Ti6Al4V alloy for both lubricants, water and PAO6 oil. In water, the coefficient of friction is reduced from 0.4 down to 0.35 (13%), and the sliding direction had almost no effect on friction. In PAO6, the coefficient of friction is reduced from 0.45 down to 0.37 (18%), which is strongly related to texturing density and sliding direction. The addition of MoS_2_ nanotubes into water and PAO additionally reduced friction of laser-textured surfaces where dimples acted as MoS_2_ nanotubes reservoirs providing a permanent friction-reducing effect. In water and in PAO, the lowest and most stable coefficient of friction of around 0.15 was observed for the highest texturing density with the smallest dimples, 50–100. 

Besides friction, wear was also studied. We found that the water lubrication in general results in higher wear compared to PAO6 oil, with the differences being exaggerated as the texturing pattern density is increased. In water and PAO6, an addition of MoS_2_ nanotubes decreased wear practically on all textured surfaces, while the one with the lowest texturing density (50–100) provides the lowest friction. 

The best results regarding friction and wear were found on the 100–200 surface lubricated by PAO oil with 2 wt.% MoS_2_, where low wear rates of the untextured surface were maintained while friction was reduced by more than 30%. For water, the best tribological results were obtained for the 100–400 pattern in terms of wear and 50–100 in terms of friction upon the addition of 2 wt.% MoS_2_.

The current research confirms that water lubrication is competitive with oil lubrication for specific laser-textured configurations of Ti6Al4V surface, especially upon the addition of MoS_2_ nanotubes as a solid lubricant. This suggests that laser-textured Ti6Al4V alloy can also be used in tribologically challenging environments using water as a lubricant and promote the concept of green tribology with minimal environmental and biological impact.

## Figures and Tables

**Figure 1 materials-15-02974-f001:**
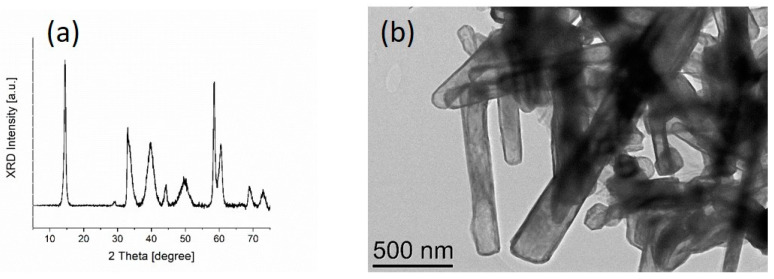
MoS_2_ nanotubes: (**a**) XRD spectrum; (**b**) TEM image.

**Figure 2 materials-15-02974-f002:**
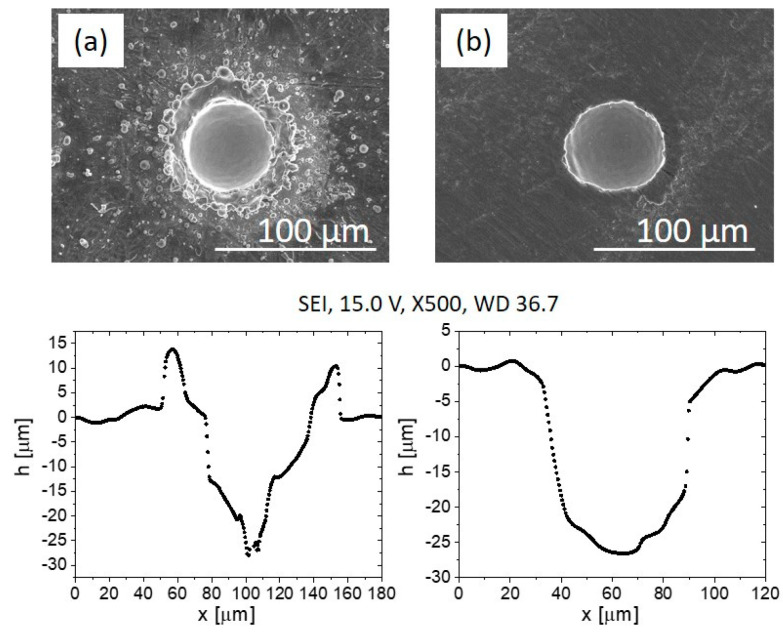
Scanning electron microscope (SEM) images of dimple textured surface (50–200: dimples with diameter 50 µm and centre-to-centre distance 200 µm) and the corresponding dimples’ profiles immediately after laser texturing (**a**) and after grinding (**b**).

**Figure 3 materials-15-02974-f003:**
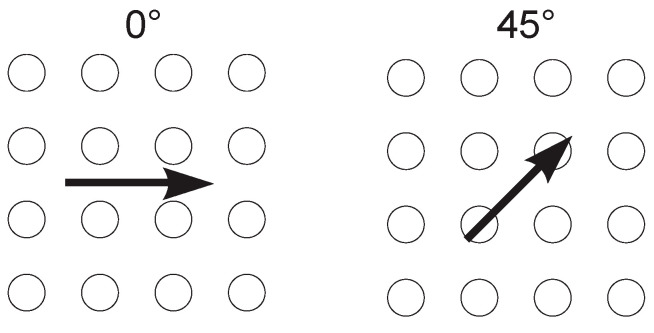
Sliding orientation.

**Figure 4 materials-15-02974-f004:**
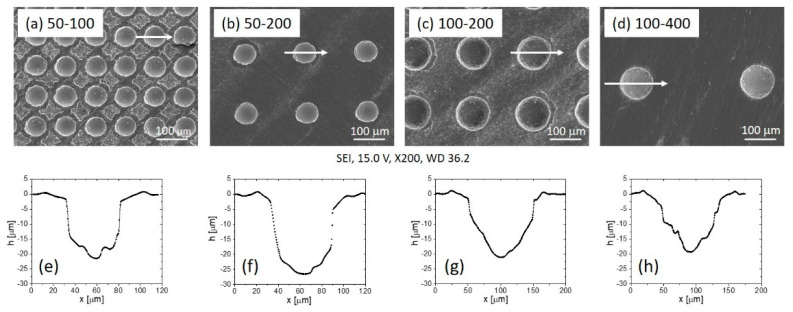
SEM images of the laser-textured Ti6Al4V surfaces with labelled diameter of the dimples (first number) and centre-to-centre distance in µm (second number): (**a**) 50–100, (**b**) 50–200, (**c**) 100–200, (**d**) 100–400, (**e**–**h**) corresponding depth profiles. The arrows on the SEM images indicate the direction of the dimple’s profile measurement.

**Figure 5 materials-15-02974-f005:**
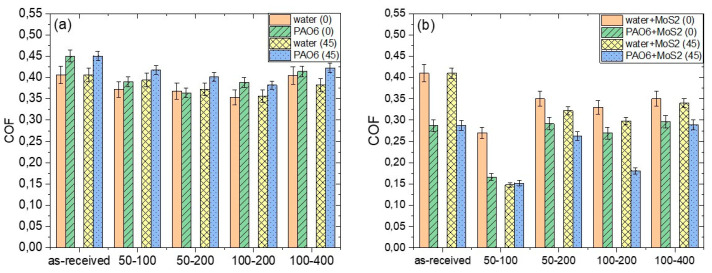
Average steady-state coefficient of friction for textured Ti6Al4V surface under water and PAO6 oil lubrication; (**a**) without and (**b**) with addition of 2 wt.% of MoS_2_ nanotubes.

**Figure 6 materials-15-02974-f006:**
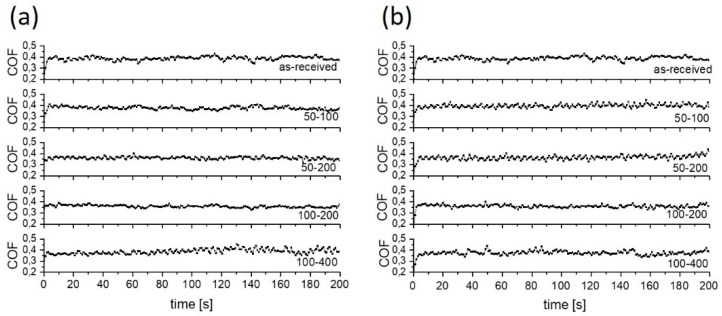
Typical coefficient of friction curves for water lubrication; (**a**) sliding along the texturing pattern and (**b**) at 45° angle.

**Figure 7 materials-15-02974-f007:**
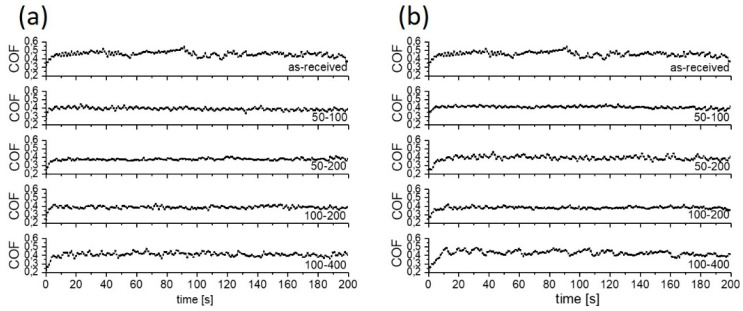
Typical coefficient of friction curves for PAO 6 lubrication; (**a**) sliding along the texturing pattern and (**b**) at a 45° angle.

**Figure 8 materials-15-02974-f008:**
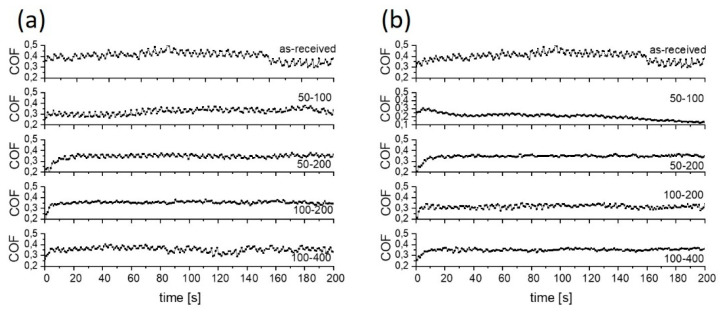
Typical coefficient of friction curves for water with 2 wt.% of MoS_2_ lubricant; (**a**) sliding along the texturing pattern and (**b**) at 45° angle.

**Figure 9 materials-15-02974-f009:**
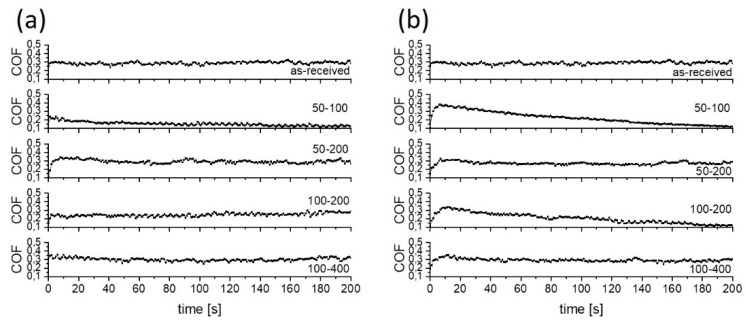
Typical coefficient of friction curves for PAO 6+2 wt.% of MoS_2_ lubricant; (**a**) sliding along the texturing pattern and (**b**) at 45° angle.

**Figure 10 materials-15-02974-f010:**
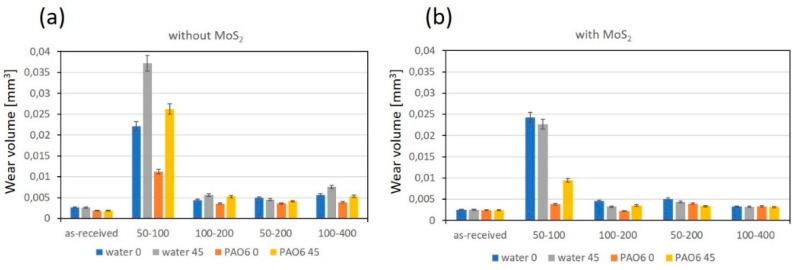
Wear volumes for non-textured and textured Ti6Al4V disc under water and PAO6 lubrication; (**a**) without and (**b**) with addition of MoS_2_ nanotubes.

**Figure 11 materials-15-02974-f011:**
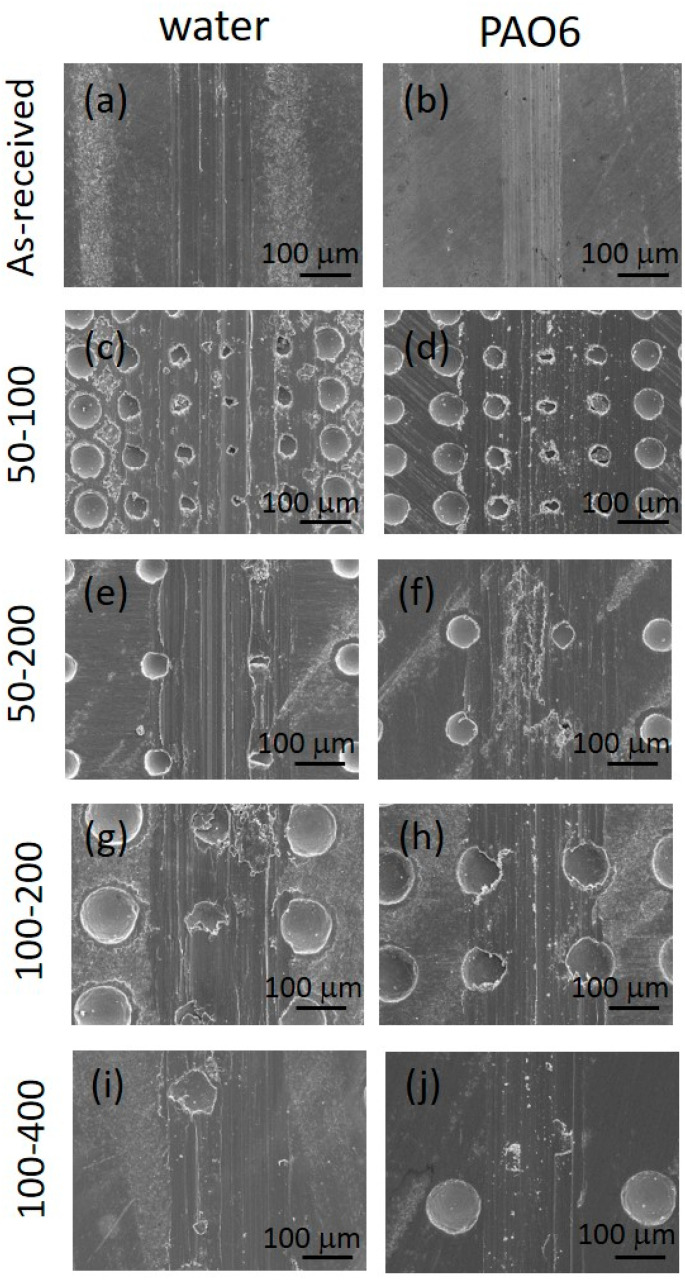
Wear scar SEM micrographs in water and PAO 6 lubrication for as-received (**a**,**b**) and laser-textured Ti6Al4V samples: 50–100 (**c**,**d**), 50–200 (**e**,**f**), 100–200 (**g**,**h**) and 100–400 (**i**,**j**).

**Figure 12 materials-15-02974-f012:**
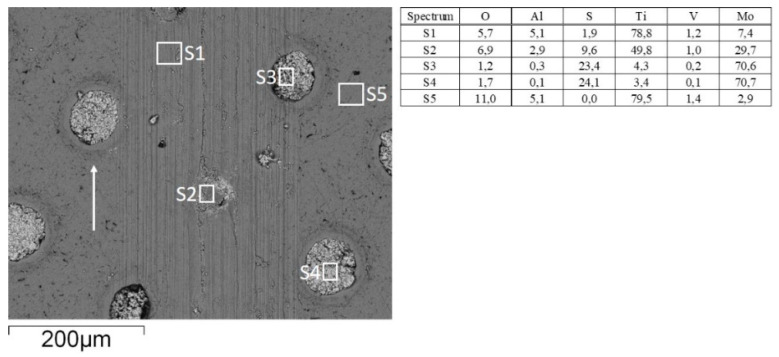
MoS_2_ nanotubes agglomeration and filling up the dimples (water+MoS_2_ lubrication; 50–200 at 45°); arrow indicates the sliding direction. Data from EDS analysis performed in five points S1–S5 are shown in wt.%.

**Figure 13 materials-15-02974-f013:**
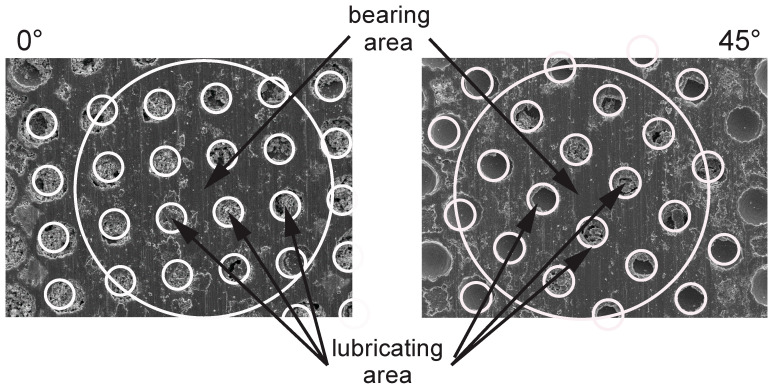
Contact area for 0° and 45° sliding.

**Figure 14 materials-15-02974-f014:**
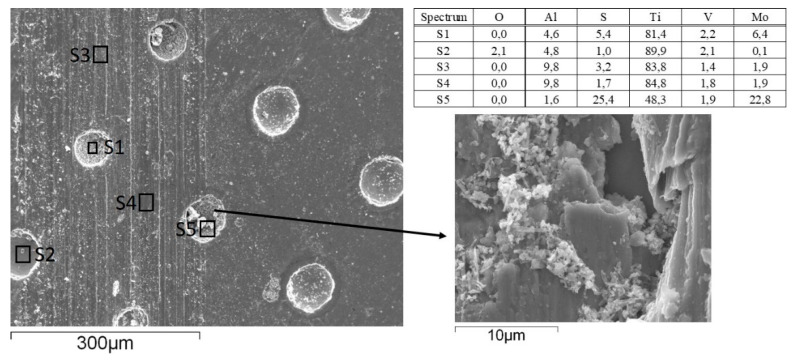
SEM micrograph of dimples after sliding under PAO6 +2 wt.% of MoS_2_ nanotubes lubrication. Data from EDS analysis performed in five points S1–S5 are shown in wt.%.

## Data Availability

Not applicable.

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
