# Peer review of "Water versus Oil Lubrication of Laser-Textured Ti6Al4V Alloy upon Addition of MoS_2_ Nanotubes for Green Tribology"

_materials, 2022, doi:10.3390/ma15092974_

Round 1

Reviewer 1 Report

In the paper the authors study the tribological behaviour of same un-textured of laser textured samples, having water or oil lubrication, with or without addition of MoS2 nanotubes. The main goal of the paper is the to evaluate the friction and wear in water and PAO6 of the laser-textured configurations, a topic of great importance for the actual science and technology. Moreover, some aspects concerning the wear are also presented and discussed. It is proved that, in certain conditions, the water may be used as lubricant.

The paper is addressed to a large audience including researchers, scholars and PhD students. The references are actual (since 2000, only one reference being from 1994) and their critical analysis is well conducted and presented. The experiences are correctly realized, the authors offering details concerning the material used in the experiments, surface laser texturing, surface characterization, tribological testing. The results include surface morphology, coefficient of friction and wear. Each conclusion is sustained by the experimental results presented inside the paper.

The paper is well written and easy to read.

In my opinion, the paper can be published in the current form.

Author Response

Thank you for the comment.

Author Response

  1. Include the physical and mechanical properties of Ti-6Al-4V titanium alloy.

Thank you for the comment, we included the physical characteristics provided by the manufacturer and the hardness of the material.

  1. In the entire manuscript maintain common notation MoS2 or MoS2

Thank you for the comment. The text was corrected.

  1. Include the wear volume formula.

Thank you for the comment. Wear volume was not calculated but directly measured with 3D optical microscope, substracting original textured surface from the wear track. Statement is corrected (Page 4):

»Each test was repeated at least three times and average steady-state coefficient of friction (last 50 seconds) calculated and wear volume measured with optical 3D measurement system Alicona InfiniteFocus G4, subtracting original textured surface from the wear track.«

  1. Refer latest articles related to Ti-6Al-4V. Surya, M. S., Prasanthi, G., Kumar, A. K.,

Sridhar, V. K., & Gugulothu, S. K. (2021). Optimization of cutting parameters while

turning Ti-6Al-4 V using response surface methodology and machine learning

technique. International Journal on Interactive Design and Manufacturing

(IJIDeM), 15(4), 453-462.

Thank you for the comment, the reference was implemented in the text.

  1. The paper should be proofread again, preferably by a technically qualified person or a

professional technical editor.

Thank you for the valuable comment. The proofreading will be performed.

  1. In conclusions, it should not be. In our study, we have shown,…….. It should be This

study shows……..

Thank you for the  comment. The text was corrected.

  1. The COF and Wear volume values need to be tabulated

Thank you for the valuable comment. In our opinion, here, graphical presentation is more representative as trend can be easily seen. In addition, graphs are also accompanied with error bars.

Reviewer 3 Report

Surface texturing was made on the Ti6Al4V sample with different parameters. Friction and wear tests were conducted to analysis and compare the behaviours of un-textured and textured samples under water and oil lubrication without and with the the addition of MoS2 nanotubes. The results proved that MoS2 nanotubes had positive effect on friction in both media for textured Ti6Al4V. The obtained results are explained in details. This paper has reasonable structure and sufficient data. But some improvements are required before this manuscript can be considered for publication:

  1. Friction and wear tests were performed at a normal load of 5 N, sliding speed of 5 mm/s, and total sliding time of 200 sec. The test parameters are too small, which are significantly lower than those of the actual application. Why choose the friction parameters like this.
  2. If the wear morphology and wear volume of the 100Cr6 counter-ball are provided, it will be more conducive to the research.
  3. Reference part should cite more relevant journal articles published in the last 3-5 years.

Author Response

Surface texturing was made on the Ti6Al4V sample with different parameters. Friction and wear tests were conducted to analysis and compare the behaviours of un-textured and textured samples under water and oil lubrication without and with the the addition of MoS2 nanotubes. The results proved that MoS2 nanotubes had positive effect on friction in both media for textured Ti6Al4V. The obtained results are explained in details. This paper has reasonable structure and sufficient data. But some improvements are required before this manuscript can be considered for publication:

  1. Friction and wear tests were performed at a normal load of 5 N, sliding speed of 5 mm/s, and total sliding time of 200 sec. The test parameters are too small, which are significantly lower than those of the actual application. Why choose the friction parameters like this.

Thank you for the valuable comment. In this investigation we focused on medium-load (600 MPa), low-sliding speed conditions, being very critical in terms of boundary lubrication. It is true 200 sec of sliding is very short, but we mainly aimed at studying the running-in behaviour. Reason behind selecting these specific conditions is included in the manuscript (Page 4):

»Friction and wear tests were performed at a normal load of 5 N, corresponding to the nominal contact pressure of 600 MPa, average sliding speed of 5 mm/s (frequency of 0.25 Hz and stroke of 10 mm) and total sliding time of 200 sec (sliding distance of 1 m). These specific contact conditions were selected in order to simulate medium-load, low sliding speed conditions, being critical in terms of boundary lubrication. Furthermore, main focus was on initial running-in performance. 100Cr6 ball with a diameter of 10 mm was used as a counter-body, load...« 

  1. If the wear morphology and wear volume of the 100Cr6 counter-ball are provided, it will be more conducive to the research.

Thank you for the comment 100Cr6 ball, having much higher hardness than the Ti-alloy (750 HV vs. 340 HV) was selected in order to concentrate wear on the Ti-alloy disc. When analyzing »wear scars« of the ball only very slight sliding marks could be observed (wear volume was within measurement error), regardless of the texturing and lubricant used. This is now indicated in manuscript (Page 8):

»In terms of wear, no measurable wear could be detected on 100Cr6 ball, being more than 2-times harder than the Ti6Al4V disc, regardless of the texturing pattern and lubricant used. However, for Ti6Al4V disc a water lubrication...«

  1. Reference part should cite more relevant journal articles published in the last 3-5 years.

Thank you for the comment. Some newer references were added.

Round 2

Reviewer 2 Report

Paper is accepted for publication